# Thermal and electrostatic tuning of surface phonon-polaritons in LaAlO₃/SrTiO₃ heterostructures

Yixi Zhou[1,2], Adrien Waelchli[1], Margherita Boselli[1], Iris Crassee[1], Adrien Bercher[1], Weiwei Luo [1,3], Jiahua Duan[4], J.L.M. van Mechelen[5], Dirk van der Marel [1], Jérémie Teyssier [1], Carl Willem Rischau[1], Lukas Korosec [1], Stefano Gariglio [1], Jean-Marc Triscone[1] & Alexey B. Kuzmenko [1] ✉

Phonon polaritons are promising for infrared applications due to a strong light-matter coupling and subwavelength energy confinement they offer. Yet, the spectral narrowness of the phonon bands and difficulty to tune the phonon polariton properties hinder further progress in this field. SrTiO₃ – a prototype perovskite oxide - has recently attracted attention due to two prominent far-infrared phonon polaritons bands, albeit without any tuning reported so far. Here we show, using cryogenic infrared near-field microscopy, that long-propagating surface phonon polaritons are present both in bare SrTiO₃ and in LaAlO₃/SrTiO₃ heterostructures hosting a two-dimensional electron gas. The presence of the two-dimensional electron gas increases dramatically the thermal variation of the upper limit of the surface phonon polariton band due to temperature dependent polaronic screening of the surface charge carriers. Furthermore, we demonstrate a tunability of the upper surface phonon polariton frequency in LaAlO₃/SrTiO₃ via electrostatic gating. Our results suggest that oxide interfaces are a new platform bridging unconventional electronics and long-wavelength nanophotonics.

Phonon polaritons (PhPs)[1–3] - hybrid modes involving light and ionic vibrations – allow spatial squeezing of the electromagnetic energy[4–8] thus facilitating applications in molecular sensing[9,10], thermal management[11,12], sub-diffraction imaging[13–17] and other technologies. However, PhPs only emerge within narrow spectral windows - Reststrahlen bands - above the transverse optical phonon frequency $\omega_{TO}$, where the real part of the material permittivity $\varepsilon(\omega)$ is negative[1]. In the case of propagating surface phonon-polaritons (SPhPs), the upper limit is set by the surface optical phonon frequency $\omega_{SO}$, where $\text{Re}\left[\varepsilon(\omega_{SO})\right] = -1$. As compared to theory, practically relevant SPhP windows are significantly more narrow when reasonable figures of

merit are set for applications[1,18]. The palette of established PhP materials does not cover the entire infrared and terahertz ranges, motivating the search of new PhP-supporting media. Another bottleneck is the difficulty to modulate the PhP properties in situ since they are dictated by the intrinsic crystal-lattice dynamics and the device geometry. Although a certain degree of dynamic SPhP tuning via coupling to electrostatically controlled graphene plasmons[19], atomic intercalation[20,21] and photoinjection[22] has been demonstrated, this direction of nanophotonics is still in its infancy.

Strontium titanate (SrTiO₃) – a prototype perovskite commonly used as a substrate in oxide electronics - has been recently suggested

[1]Department of Quantum Matter Physics, University of Geneva, CH–1211, Geneva 4, Switzerland. [2]Beijing Key Laboratory of Nano-Photonics and Nano-Structure (NPNS), Department of Physics, Capital Normal University, 100048 Beijing, China. [3]The Key Laboratory of Weak-Light Nonlinear Photonics, Ministry of Education, School of Physics and TEDA Applied Physics Institute, Nankai University, Tianjin 300457, China. [4]Department of Physics, University of Oviedo, Oviedo 33006, Spain. [5]Department of Electrical Engineering, Eindhoven University of Technology, 5600 MB, Eindhoven, Netherlands. ✉e-mail: Alexey.Kuzmenko@unige.ch

as a promising PhPs medium[18,23–27] since it features two prominent Reststrahlen bands spanning from the mid-infrared to terahertz ranges[28]. In addition, SrTiO₃ (STO) demonstrates a plethora of non-trivial physical phenomena, such as a soft-phonon mode behavior[28], mid-infrared polaronic absorption[29] and superconductivity at low carrier concentrations[30]. Of particular interest are the surface properties of the STO, specifically a formation of a conducting two-dimensional electron gas at its interface with other insulating oxides[31–35]. Importantly, the lately demonstrated fabrication of ultra-thin free-standing layers of STO and other oxide perovskites[36–38] places this family on a par with the two-dimensional van der Waals materials in the ongoing quest for subwavelength polariton confinement[27]. It is

tempting to leverage the unique features of STO to achieve a better control of the phonon polaritons and build new functionalities in nanophotonic devices.

In this article, we explore the possibility of using temperature variation and electrostatic gating to tune the spectral and propagational properties of SPhPs in pristine STO and LaAlO₃/SrTiO₃ (LAO/STO) heterostructures. To this end, we use scattering-type scanning near-field optical microscopy (s-SNOM), which has already shown its potential for mid-infrared imaging of the two-dimensional electron gas (2DEG) in LAO/STO[23,39] above the phonon bands. Here we focus on the upper Reststrahlen band of STO (Fig. 1a) born out of the optical phonon mode $\upsilon_4$ as depicted in the inset[40], and spanning from

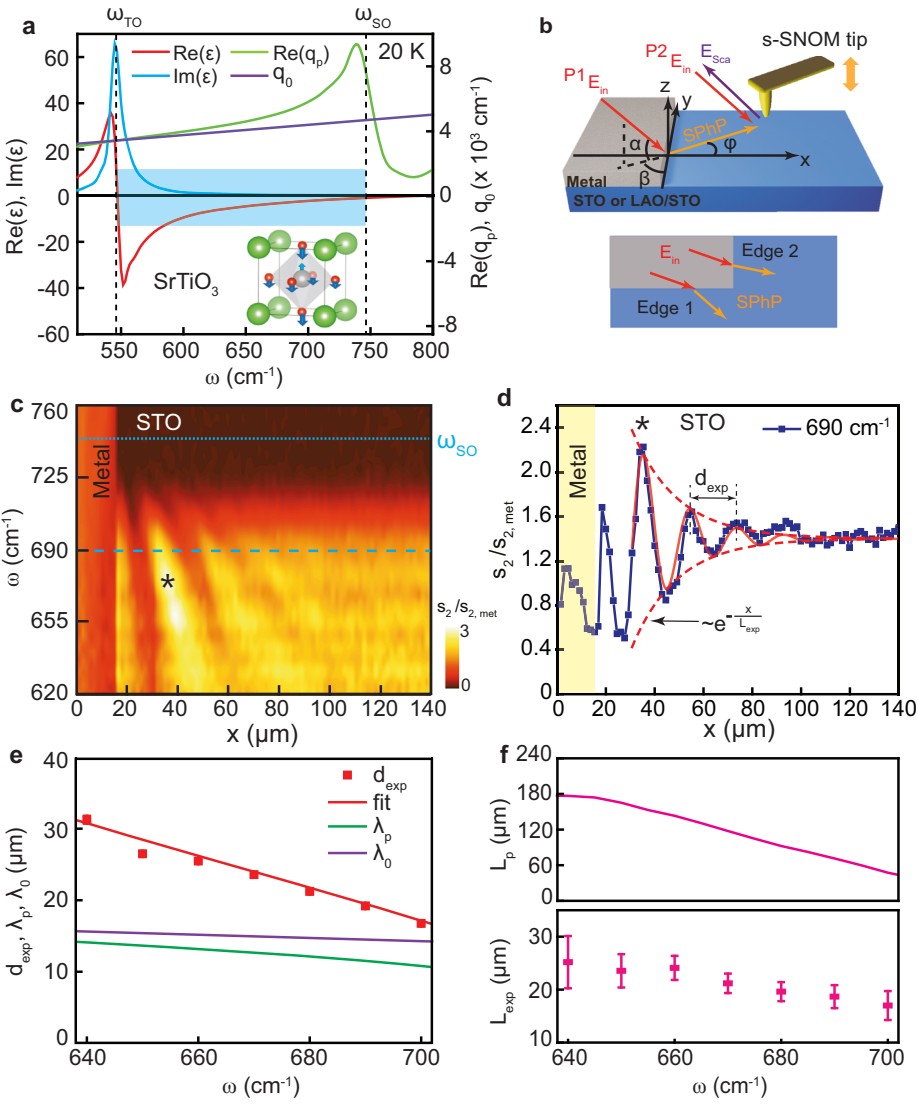

**Fig. 1 | Real-space nano-spectroscopy of surface phonon polaritons in pristine SrTiO₃ at 15 K. a** Left axis: real (red line) and imaginary (blue line) parts of the STO permittivity $\varepsilon_{STO}(\omega)$ in the upper SPhP band (shaded area) corresponding to the highest-frequency phonon mode $\upsilon_4$ (depicted in the inset). Right axis: the real part of the in plane SPhP momentum $q_p = q_0\sqrt{\varepsilon_{STO}/(1+\varepsilon_{STO})}$ (green line) and the momentum of light in free space $q_0 = \omega/c$ (purple line). The inset shows the unit cell of STO (Sr – green, Ti – gray, O – red), where the arrows denote the vibration eigen vector of the LO mode according to ref. 40. **b** Schematics of the SPhP interferometry experiment. The incident infrared beam $E_{in}$ (red arrows) illuminates both the metal edge and the s-SNOM tip. The edge launched SPhPs (orange arrows) propagate toward the tip. The back-scattered radiation $E_{sca}$ contains contributions from two paths: the one mediated by the SPhPs (P1) and the one directly scattered by the tip (P2). As a result, an interference pattern is formed as a function of the

edge-tip distance. $\alpha$, $\beta$ and $\varphi$ represent the incident angle, the azimuthal angle and SPhP propagating angle, respectively. **c** Nano-FTIR distance-frequency map of the s-SNOM amplitude demodulated at the 2ⁿᵈ tapping harmonics and normalized by the signal on metal. Light blue dashed line represents the surface optical phonon frequency $\omega_{SO}$. **d** Symbols: line profile at 690 cm⁻¹ (dashed dark-blue line in c). Solid line: a damped-sine function fit of the data. **e** Symbols: the fringe spacing $d_{exp}$ extracted from the fits, as a function of frequency. Red solid line: the best fit using formula (1) as described in the text. Green and purples lines: the SPhP wavelength $\lambda_p = 2\pi/\text{Re}(q_p)$ and the light wavelength $\lambda_0$ respectively. **f** Calculated propagation length $L_p = 1/\text{Im}(q_p)$ (top panel) and experimental SPhP decay length $L_{exp}$ (bottom panel) as a function of frequency. All error bars correspond to the standard deviation.

$\omega_{\text{TO}} = 546 \, \text{cm}^{-1}$ to $\omega_{\text{SO}} = 746 \, \text{cm}^{-1}$ (at 20 K)[29,41,42]. The frequency dependent dielectric function of STO, $\varepsilon_{\text{STO}}(\omega)$, is well known[18,28,29,41,42] and we use it as a reference for interpretation of our data. Figure 1a presents the real (red line) and imaginary (blue line) parts of $\varepsilon_{\text{STO}}(\omega)$ at 20 K. Under the assumption that the dielectric function near the surface is the same as in the bulk, the complex-valued in-plane SPhP wavevector in pristine STO can be obtained from the standard relation[1] $q_{\text{p}}(\omega) = q_0(\omega)\sqrt{\frac{\varepsilon_{\text{STO}}(\omega)}{1+\varepsilon_{\text{STO}}(\omega)}}$, where $q_0(\omega) = \omega/c = 2\pi/\lambda_0$ is the wavevector of light ($c$ is the light velocity). As it is typical for surface modes in semi-infinite samples, $\text{Re}(q_{\text{p}})$ (green line) is only slightly larger than $q_0$ (purple line) indicating a weak level of confinement[43].

## Results

### Experimental technique

To image SPhPs in a broad range of temperatures and frequencies, we use a cryogenic (6–300 K) s-SNOM platform equipped with a nanoscale Fourier transform infrared (nano-FTIR) module allowing us to obtain continuous spectra down to 600 cm⁻¹. Developing upon the original method applied to SiC[44], we use the experimental geometry depicted in Fig. 1b. A thick STO or LAO/STO sample is partly covered by a 30 nm thick layer of Au or Pt. The sample and a metal-coated s-SNOM tip are simultaneously illuminated with an infrared laser. The straight and sharp metal edge converts the incident field ($E_{\text{in}}$) into near fields providing the necessary in-plane momentum to launch the SPhPs. The field scattered by the tip ($E_{\text{sca}}$) is captured with a remote infrared detector and demodulated at several harmonics of the tip tapping frequency. In this work we focus on the second and third harmonics of the s-SNOM amplitude, $s_2$ and $s_3$. When the tip is sufficiently close to the edge, two paths (P1 and P2) of scattering are active. In P1, the edge launched SPhPs propagate along the surface towards the tip and scatter out to free space, while in P2 the incident beam is directly backscattered at the tip. The fields from P1 and P2 interfere constructively or destructively according to their phase difference, which depends on the tip-edge distance and the illumination direction as specified below. We note that an alternative path, where the tip-launched SPhPs are reflected at the edge and scattered by the tip, can be neglected in the case of weakly confined surface polaritons[45,46].

In our samples, two orthogonal metal edges (1 and 2) are used, allowing us to compare two illumination geometries (Fig. 1b and Supplementary Fig. S1). The projection of the incident-light momentum to the sample surface is nearly parallel to edge 1 and, correspondingly, almost orthogonal to edge 2. In all cases, the tip is scanned along a line perpendicular to the edge ($x$-axis). In Fig. 1c, we present the hyperspectral $x$-$\omega$ map of the s-SNOM amplitude $s_2$ obtained on edge 1 on top of pristine STO at 15 K and normalized to the signal on metal, $s_{2,\text{met}}$ (the map on edge 2 is shown in Supplementary Fig. S2a). Below approximately 725 cm⁻¹, a series of dispersive fringes are observed on the sample near the metal, which decay as the tip moves away from the edge. To extract the fringe spacing and the decay length, we fit the spatial profiles $s_2(x)/s_{2,\text{met}}$ separately at each frequency with a damped-sine function $A e^{-\frac{x}{L_{\text{exp}}}} \cdot \sin(2\pi \frac{x}{d_{\text{exp}}} - \varphi_0)$, where the amplitude $A$, phase $\varphi_0$, fringe spacing $d_{\text{exp}}$ and decay length $L_{\text{exp}}$ are adjustable parameters. In Fig. 1d and Supplementary Fig. S2b, the experimental profiles and the fits at 690 cm⁻¹ are presented for edges 1 and 2 respectively. Figure 1e, f show $d_{\text{exp}}$ and $L_{\text{exp}}$ (symbols) as a function of $\omega$ for edge 1 (Supplementary Fig. S2c, d for edge 2). Notably, the fringe period is longer than the calculated phonon polariton wavelength $\lambda_{\text{p}} = 2\pi/\text{Re}(q_{\text{p}})$ (green line in Fig. 1e). Moreover, the fringe distance for edge 2 is larger than for edge 1. Both observations are in contrast to the case of edge-launched ultraconfined ($\lambda_{\text{p}} \ll \lambda_0$) PhPs in van der Waals crystals, where $d_{\text{exp}} = \lambda_{\text{p}}$ and is edge-independent[47,48]. However, this is expected in our case since $\lambda_{\text{p}} \sim \lambda_0$ (purple line in Fig. 1e

Supplementary Fig. S2c)[43,44]. As shown in the Supplementary Note S3, the fringe spacing for an arbitrary illumination direction is given by the formula:

$$d = \lambda_{\text{p}} \cos\varphi \left[ 1 - \frac{\lambda_{\text{p}}}{\lambda_0} \cos\alpha \sin(\varphi - \beta) \right]^{-1} \quad (1)$$

where $\alpha$ and $\beta$ are the incident and azimuthal illumination angles respectively and $\varphi$ is the angle between the SPhP momentum and the $x$-axis (Fig. 1b and Supplementary Fig. S3). The SPhP angle, in turn, is determined by the surface-polariton analog of the optical Snell's law:

$$\sin\varphi = \frac{\lambda_p}{\lambda_0} \cos\alpha \cos\beta. \quad (2)$$

The illumination angles are highly sensitive to the alignment of the laser-focusing mirror and hard to measure in the cryo-SNOM setup. To circumvent this problem, we model $d_{\text{exp}}$ simultaneously for both edges with Eqs. (1) and (2), while treating $\alpha$ and $\beta$ as adjustable parameters and keeping in mind that $\alpha$ on the two edges is the same, but the values of $\beta$ differ by 90°. Using this procedure, we obtain a good match between the experiment and theory (red solid lines in Fig. 1e and Supplementary Fig. S2c) for the values of $\alpha = 23.3°$ and $\beta = 7.8°$ for edge 1 (97.8° for edge 2). The possibility to reproduce $d_{\text{exp}}(\omega)$ at many frequencies and for both edges with only two parameters corroborates that the fringe patterns are governed by the SPhP interference and not by optical artefacts unrelated to the sample properties.

The propagation length $L_p = 1/\text{Im}(q_{\text{p}})$ is a sensitive measure of the phonon scattering and is an essential figure of merit for applications[1]. As clearly seen in Fig. 1c and Supplementary Fig. S2a the interference persists at about hundred microns from the edge (80 μm for edge 1 and 140 μm for edge 2). This tells unequivocally that the SPhPs travel in our sample by at least the same distance before they become indistinguishable from noise (a similar observation at room temperature is made in ref. 26). We note that the maximum extent of the SPhP interference observed in SiC – a textbook PhP material - is about the same[44,49]. And yet, the theory based on the bulk dielectric function of STO (Fig. 1a) predicts a several times longer propagation length $L_{\text{p}}$ than the experimental decay length $L_{\text{exp}}$ (Fig. 1f). A possible reason for this difference is the presence of additional electromagnetic losses on the surface due to oxygen vacancies or other defects. However, we believe that in the present case the decay length obtained by the damped-sine function fitting is artificially suppressed because of the limited size of the illumination spot[43] and can therefore serve only a lower bound for the true SPhP propagation length.

### Thermal tuning of SPhPs in STO

To investigate the thermal variation of SPhP properties in bare STO, we analyze nano-FTIR profiles measured along the same path at various temperatures $T$ from 15 to 300 K. Figure 2a presents the s-SNOM amplitude (symbols) and the corresponding damped-sine function fits (lines) at 690 cm⁻¹ (the data at other frequencies show a similar trend). One can notice that the temperature variation of the amplitude profiles is rather small. Accordingly, the extracted values of $d_{\text{exp}}$ and $L_{\text{exp}}$ (purple and orange symbols in Fig. 2b) show no or a weak temperature dependence. For comparison, we present on the same graph the calculated fringe spacing and the propagation length based on the dielectric function of STO at different temperatures (Fig. 2c)[42]. The real part of $\varepsilon_{\text{STO}}(\omega)$ is almost temperature independent in this frequency range, resulting in a rather weak variation of $d$, corroborating the observations. In contrast, $\text{Im}[\varepsilon_{\text{STO}}(\omega)]$ strongly decreases with temperature due to a reduction of the phonon damping[50], which entails a strong increase of the theoretical value of $L_{\text{p}}$ with cooling down not observed in our measurements. This difference can be naturally

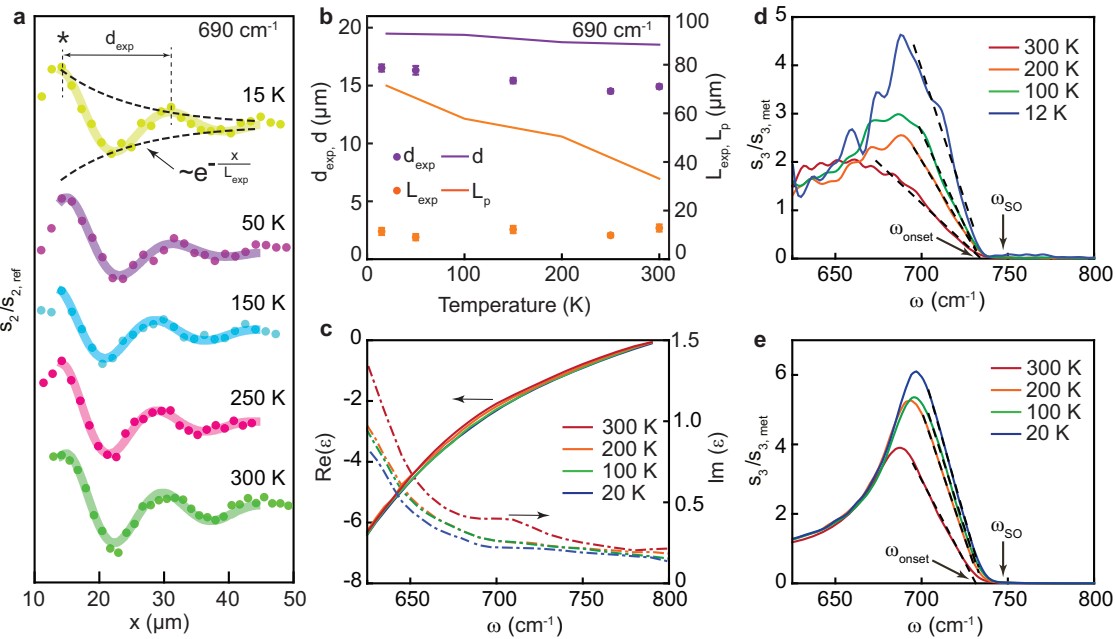

**Fig. 2 | Temperature dependence of the SPhP properties in pristine SrTiO₃.**
**a** Symbols: s-SNOM amplitude profiles at 690 cm⁻¹ at different temperatures from 15 to 300 K; solid lines: the corresponding damped-sine function fits. The curves are vertically shifted for clarity. The asterisk denotes the most intense maximum also marked in Fig. 1c, d. **b** Extracted fringe spacing $d_{exp}$ (purple) and decay length $L_{exp}$ (orange) as a function of temperature. Solid lines: calculated temperature dependence based on the dielectric function of STO (panel **c**). A small difference between the experimental and theoretical values of $d$ is likely due to non-identical

illumination conditions in this measurement as compared to Fig. 1c, which we used to set the angles $\alpha$ and $\beta$ in the calculation as specified in the text. **c** Real (solid curve) and imaginary (dash curve) parts of $\varepsilon_{STO}(\omega)$ at different temperatures[42]. **d** Nano-FTIR amplitude spectra $s_3/s_{3,met}$ collected far away from the metal edge at 12, 100, 200 and 300 K. **e** s-SNOM amplitude spectra obtained via the finite-dipole model using the permittivity data from panel **c**. Dashed lines in **d** and **e**: linear fits of the right-side part of the SPhP peak used to extract the onset SPhP frequency $\omega_{onset}$ as described in the text. All error bars correspond to the standard deviation.

explained by the mentioned effect of the finite spot size which limits the decay length and masks its true temperature dependence.

Having established the existence of long-propagating SPhPs in STO in a broad range of temperatures, we focus on the frequency dependence of the s-SNOM signal. In Fig. 2d, we present the spectra of $s_3(\omega)/s_{3,met}$ at 12, 100, 200 and 300 K collected 150 μm away from the metal edge, where the SPhP interference is absent. The signal increases abruptly below a certain onset frequency ~735 cm⁻¹ and forms a pronounced peak at ~700 cm⁻¹. Since the onset frequency is close to $\omega_{SO}$ (indicated by arrow) and is just above the maximum frequency where SPhP fringes are observed, we assign the steep signal rise to the generation of SPhPs by the tip. Interestingly, the onset frequency barely changes with temperature while the peak intensity grows with cooling down. To understand this, we calculate the s-SNOM spectra (Fig. 2e) at various temperatures via the finite-dipole model of the sample-tip interaction[51,52] (See Methods and Supplementary Note S4) using the dielectric function of STO shown in Fig. 2c. The remarkable agreement between the trends seen in the experimental and theoretical spectra indicates that the onset frequency is indeed linked to $\omega_{SO}$, which is in turn determined by Re($\varepsilon_{STO}$) and is therefore weakly $T$-dependent. On the other hand, the peak intensity is related to the optical losses, controlled by Im($\varepsilon_{STO}$), and is higher at low temperatures. We note that the physical meaning of the peak frequency and the interpretation of its temperature dependence are less clear than the ones of the onset frequency.

## Thermal tuning of SPhPs in LAO/STO

We move now to the discussion of the experiments performed on conducting LAO/STO heterostructures. The LAO thickness of our sample, 6 unit cells (about 2 nm), is above the threshold for the formation of the 2DEG at the interface with the STO substrate[32]. First, we notice that hyperspectral maps (Supplementary Fig. S5) clearly show interference fringes similar to the ones in pristine STO (Fig. 1c) telling

us that long-propagating SPhPs persist in the presence of the 2DEG. Next, we focus on the analysis of the nano-FTIR spectra collected far away from the edge. In Fig. 3a, the spectra of $s_3(\omega)/s_{3,met}$ at 12, 100, 200, and 300 K are shown. As it is the case in bare STO (Fig. 2d), the s-SNOM amplitude grows quickly below a certain frequency. However, in contrast with pristine STO, the onset frequency now shows a strong blueshift as the temperature is decreased. To quantitatively compare the thermal shifts of the phonon structure in the two systems, we adopt a common ad hoc definition of the onset frequency, $\omega_{onset}$. We determine it as the intersection of the straight line fitting the high-frequency slope of the phonon peak with the horizontal axis as sketched in Figs. 2d and 3a. Since this fit involves a broad range of frequencies, the relative uncertainty of the onset frequency (for example, its shift as a function of temperature) is much better than the spectral resolution of the nano-FTIR spectroscopy (see Methods). One can see that $\omega_{onset}(T)$ for LAO/STO (red downward triangles in Fig. 3d) blue-shifts by about 14 cm⁻¹ between room temperature and 12 K, whereas it changes by less than 2 cm⁻¹ in pristine STO (black circles). This striking difference is likely caused by the 2DEG, since the thin insulating LAO layer cannot cause such a strong effect (Supplementary Note S6). To corroborate this hypothesis, we perform a finite dipole simulation of the s-SNOM spectra of the multilayer LAO/2DEG/STO system at different temperatures. We set the thickness $t$ of the 2DEG layer to 2.7 nm in agreement with previous studies[53,54]. While the presence of the 2DEG can be detected using infrared ellipsometry[54,55], no spectra of $\varepsilon_{2DEG}(\omega)$ were reported, because the ultrathin conduction layer contributes weakly to the far-field optical properties. Instead, despite the subtle differences in the band structures of 2DEG and chemically doped STO (as detailed in Supplementary Note S7), we set the dielectric function of the 2DEG to be the same as in Nb doped strontium titanate with a comparable level of the charge carrier concentration. Specifically, we use the dielectric function $\varepsilon_{STNO}(\omega)$ of $SrTi_{1-x}Nb_xO_3$ ($x = 0.02$)[29] at 20, 100, 200 and 300 K (Fig. 3b). At this doping, the volume carrier density

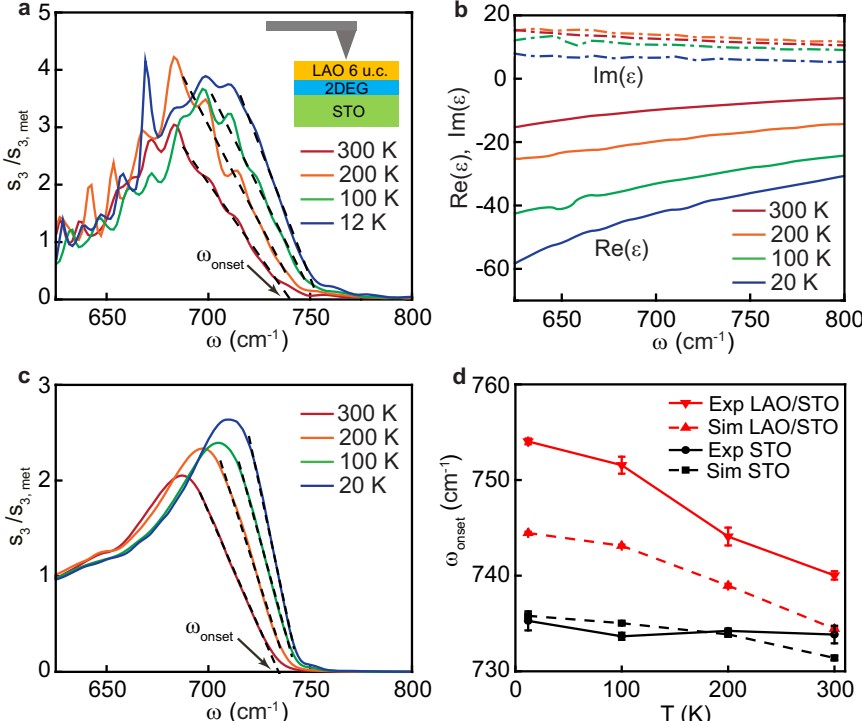

**Fig. 3 | Temperature dependence of the SPhP band in LaAlO₃/ SrTiO₃. a** Nano-FTIR spectra at 12, 100, 200 and 300 K collected far away from the metal edge. **b** Permittivity of SrTi$_{0.98}$Nb$_{0.02}$O$_3$, at different temperatures[29], which are used mimic the permittivity of the 2DEG layer. **c** Simulated s-SNOM amplitude spectra of the layered system with 2DEG, obtained using the finite-dipole model. Dashed lines in **a** and **c** are the linear fits to the high-frequency slope of the SPhP peak used to determine the onset frequency $\omega_{onset}$. **d** The onset frequency as a function of temperature. Black circles and squares are respectively experimental and model values on pristine STO, extracted from Fig. 2d, e. Red downward and upward triangles are the experimental and simulated values obtained on the LAO/STO system. All error bars correspond to the standard deviation.

is $N_{3D} = 3.4 \times 10^{20}$ cm$^{-3}$ corresponds to the 2D carrier density $N_{2D} = N_{3D}t = 9.2 \times 10^{13}$ cm$^{-2}$, which matches the carrier density in the LAO/STO system extracted from the transport Hall measurements[23]. Notably, the real part of $\varepsilon_{STNO}$ changes dramatically as a function of temperature, in a stark contrast with pristine STO. As it was shown in ref. 29, this effect is not caused by the temperature dependence of the carrier concentration (which is relatively weak) but by a polaronic screening of the charge carriers and a temperature dependent redistribution of the optical spectral weight between the low-frequency Drude peak to a mid-infrared polaron band above $\omega_{SO}$. In Fig. 3c, we present the theoretical spectra[52]. The model perfectly reproduces the most salient features of nano-FTIR measurements: the increase of the phonon peak intensity and the blueshift of $\omega_{onset}$ (red triangles in Fig. 3d) with cooling down. This suggests that the electron-phonon interaction plays an important role in the infrared response of the 2DEG.

### Electrostatic tuning of SPhPs in LAO/STO

The possibility of tuning the properties of the 2DEG is an important property of the LAO/STO heterostructures. In Fig. 4, we present the effect of back-gating of the 2DEG on the SPhP spectra. Due to the soft-mode instability – a hallmark of this material - the static permittivity of STO and the gating efficiency increase at low temperatures by several orders of magnitude, therefore we perform this experiment at 7.7 K. When the gate voltage $V_G$ is swept from 0 to −200 V, the measured in situ sheet resistance (Fig. 4a) increases by more than a factor of 6, in agreement with previous measurements on similar devices[35]. This effect is due to a decrease of charge carrier density and a concomitant reduction of the charge mobility[23,35]. On the other hand, the resistance decreases very weakly when $V_G$ is tuned from 0 to 200 V, likely due to a low gating performance at positive voltages in the SNOM setup, where

the presence of visible light radiation with the photodoping effect influences the electrostatic doping process. The s-SNOM amplitude spectra (normalized to the peak value) collected on a same point of the sample far from the metal edge at $V_G$ = 0, −50, and −150 V are shown in Fig. 4b (the spectra at other gate voltages are presented in Supplementary Fig. S7). One can see that the SPhP onset slightly redshifts when the negative voltage is applied. In Fig. 4c, we present the value of $\omega_{onset}$, obtained using the described ad hoc procedure, as a function of the gate voltage. The onset frequency drops from 743 to 738 cm$^{-1}$ as $V_G$ is swept from 0 to −200 V, but it barely changes in the positive voltage side. The curves of $\omega_{onset}(V_G)$ and $R(V_G)$ show correlated behavior indicating that the infrared SPhP frequency and DC transport properties are connected.

Below we check numerically if the gate-induced change of the dielectric function of the 2DEG close to the SPhP onset frequency (740 cm$^{-1}$) may explain this connection. Lacking direct measurements of $\varepsilon_{2DEG}(\omega)$ and especially of its variation with gating, we perform a simplified simulation. First, we assume, as before, that $\varepsilon_{2DEG}(\omega)$ at zero gate voltage is equal to $\varepsilon_{STNO}(\omega)$ and calculate $\omega_{onset,0}$ using the same ad hoc method. Then we add a constant value $\delta\varepsilon_1$ to the real part of $\varepsilon_{2DEG}$ and compute the corresponding shift $\delta\omega_{onset,1}(\delta\varepsilon_1) = \omega_{onset}(\delta\varepsilon_1, \delta\varepsilon_2 = 0) - \omega_{onset,0}$ (red symbols in Fig. 4d). Finally, we do the same calculation by adding $\delta\varepsilon_2$ to the imaginary part of $\varepsilon_{2DEG}$, and obtain $\delta\omega_{onset,2}(\delta\varepsilon_2) = \omega_{onset}(\delta\varepsilon_1 = 0, \delta\varepsilon_2) - \omega_{onset,0}$ (blue symbols). This modeling reveals that the onset frequency is affected much stronger by $\delta\varepsilon_1$ than $\delta\varepsilon_2$. Specifically, the redshifts experimentally observed at $V_G = -50$ V and −150 V (Fig. 4c) correspond in the simulation to $\delta\varepsilon_1 = +12$ and +24 respectively, as indicated by arrows in Fig. 4d. Keeping in mind that Re[$\varepsilon_{STNO}(\omega_{onset})$] ≈ −40 (Fig. 3b), these changes imply that Re[$\varepsilon_{2DEG}(\omega_{onset}, V_G = -50V)$] ≈ −28 and Re[$\varepsilon_{2DEG}(\omega_{onset}, V_G = -150V)$] ≈ −16. First, the lowering of the

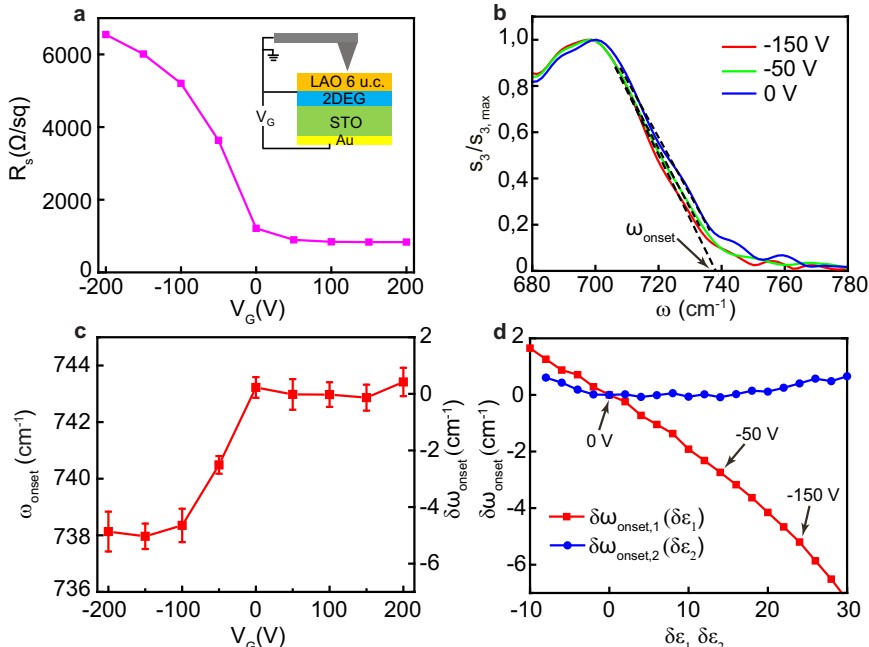

**Fig. 4 | Effect of electrostatic gating of the 2DEG on the SPhP band in LaAlO₃/STO. a** Sheet resistance of 2DEG as a function of the gate voltage, measured in situ during the cryogenic s-SNOM experiment. **b** Nano-FTIR spectra of $s_3/s_{3,max}$ (normalized to the peak value) collected far away from the metal edge at selected values of the gate voltage $V_G$: 0, −50 and −150 V. **c** The onset frequency $\omega_{onset}$ extracted using linear fits as shown in **b** and Supplementary Fig. S7 as a function of the gate voltage. The relative shift of the onset frequency with respect to $V_G$ = 0 V is given on the right axis. **d** Simulated shift of $\omega_{onset}$ obtained in two ways. First, the real part of the 2DEG permittivity is changing while the imaginary part is kept constant (red symbols). Second, the imaginary part is varied at constant real part (blue symbols). The arrows indicate the change of the real part of the permittivity corresponding to the experimentally observed shift of the onset frequency at 0 V, −50 V and −150 V. All error bars correspond to the standard deviation.

absolute value of $\varepsilon_1$ is qualitatively consistent with the expected decrease of the carrier density at negative voltages[35]. Second, the value remains negative, indicating the presence of carriers even at the highest absolute voltage applied. Therefore, we conclude that the hypothesis of the connection between the density of the carriers and the shift of the onset frequency is reasonable.

## Discussion

We have shown that the presence of the two-dimensional electron gas in the LAO/STO interface results in a stronger temperature dependence of the SPhP onset frequency, as compared to pristine STO, and it also enables a certain electrostatic control of this value: $\omega_{onset}$ blue-shifts by about 14 cm⁻¹ from 300 K to 10 K and redshifts by 5 cm⁻¹ when the maximum negative gate voltage is applied. The amount and the direction of both shifts are consistent with the expected variation of the real part of $\varepsilon_{2DEG}(\omega)$ in these processes. While the physical reason of the gate-voltage related shift of $\omega_{onset}$ is the reduction of the carrier concentration, the origin of the thermal effect is quite different, namely a redistribution of the spectral weight between the Drude peak and the polaronic mid-infrared band.

It is worth noting that applying gate voltage to the same samples does not significantly change the fringe spacing in hyperspectral $(x - \omega)$ maps (See Supplementary Note S9). This indicates that the low-momentum SPhPs, which are responsible for the interference patterns in these maps, are much less affected by the gate voltage than the high-momentum SPhPs, which determine the onset frequency in the nano-FTIR spectra. Indeed, a simulation of the phonon-plasmon polariton dispersion in LAO/STO at zero gate voltage and $V_G = -150$ V (Supplementary Note S10) shows that the low-momentum SPhPs ($q \sim q_0$, region A) are almost doping independent. The physical reason of this insensitivity is that the electromagnetic field of low-$q$ SPhPs penetrates into the STO substrate much deeper than the thickness of the 2DEG. The same calculation clearly shows a redshift of the SPhPs with the

higher momenta ($q \gg q_0$, region B), in agreement with our results. At even higher in-plane momenta $q$, an extra surface polariton branch is seen above $\omega_{onset}$ (region C), which is identified as plasmon-polariton in the 2DEG[23]. This branch demonstrates a much stronger dependence on the gate voltage than the SPhP branch. Even though the absolute s-SNOM amplitude above $\omega_{onset}$ is small (see, e.g. Fig. 3a), the relative signal change as a function of the gate voltage in this spectral range has been shown to be very significant[23].

Overall, our cryo-SNOM experiments supported by simulations suggest that strontium titanate is not only a new promising material hosting long-propagating phonon polaritons in the far-infrared range, but also a unique system where a two-dimensional electron gas, which can be easily created via heterostructuring, enables active tuning of surface PhPs, the idea initially realized in the graphene-hBN system[19]. While on weakly confined SPhPs in bulk samples, the fast progress in fabrication of ultrathin crystalline layers of STO and other perovskite oxides[36–38] will allow studying and harnessing highly confined volume waveguide PhP modes in this family of materials. We notice that the low-energy Reststrahlen band of STO, not covered by the present study, originates from a phonon mode, which softens dramatically at low temperature[28] potentially leading to a cavity-enhanced ferroelectric phase transition[56] thus realizing an important case of electrodynamic control of matter. The surface phonon polaritons from this band should be more sensitive to temperature than the ones studied here. On the experimental side, we believe that the method of two-edge interferometry developed in this work can be widely used to tackle uncertainties in the illumination geometry often present in the s-SNOM experiment. This approach is complementary to the use of circular disc launchers[57], however, it does not require a 2D SNOM mapping and therefore is easier to realize in the case of nano-FTIR spectroscopy. Furthermore, with the future improvement of theoretical models of the near-field tip-sample interaction, we expect that this technique will allow direct extraction

of the spectral optical properties of the 2DEG from the nano-FTIR spectra.

## Methods

### Sample preparation

The LAO/STO samples were grown by depositing approximately 6 unit cells of $LaAlO_3$ on commercial $TiO_2$-terminated (001)-oriented $SrTiO_3$ substrates (Crystec GmbH) by pulsed laser deposition. The growth was performed in a pure oxygen atmosphere at a pressure of $8 \cdot 10^{-5}$ Torr while the substrate is heated to 820 °C during the deposition. A post-annealing of 1 h is performed at 575 °C in an atmosphere of 0.2 bar of oxygen. The samples are then cooled down in the same oxygen pressure. After the growth a back gate electrode was deposited over the whole sample by sputtering Au on the back of the substrate. The back gate electrode was electrically contacted using silver paste to bond a Cu wire to the Au electrode allowing to apply an electric field between the latter and the conductive interface. To measure the electrical transport properties, the LAO/STO interface was contacted in a van der Pauw geometry by ultrasonic welding using Al-wires. For both LAO/STO and bare STO devices, the metallic electrodes were deposited using a standard photolithographic process to pattern the stripes on the sample (using MICROPOSIT S1813 photoresist, MICROPOSIT 351 developer), followed by the metallic layer deposition using electron beam evaporation. For the Pt stripes, 30 nm of Pt was deposited before removing the photoresist (using MICROPOSIT REMOVER 1165), while for the Au stripes 5 nm of Ti followed by 30 nm of Au were deposited. Both metals were deposited at room temperature in a high vacuum ($<10^{-7}$ Torr). For some samples, grounding of the metallic stripes during the SNOM measurements was achieved by pressing a tiny indium wire on top of the stipes.

### Cryogenic hyperspectral infrared nanoimaging

Hyperspectral infrared near-field nanoimaging at various temperatures was performed using a commercial cryogenic s-SNOM platform cryo-neasSNOM from Neaspec/Attocube GmbH equipped with a nano-FTIR module. Briefly, s-SNOM is based on an atomic force microscope (AFM), operating in a tapping mode (frequency ~240 kHz, peak-to-peak amplitude ~90 nm). The Pt-coated AFM with the apex radius of about 60 nm is illuminated by a continuum infrared DFG laser and the scattered light is spectrally analyzed using a Fourier-transform interferometer with the spectral resolution of 6 $cm^{-1}$. A pseudo-heterodyne demodulation of the s-SNOM signal at higher order tapping harmonics (2 and 3 in this work) is used to suppress the far-field background signal. The scanner span in the setup is 50 μm, therefore the long signal profiles, such as in Fig. 1c, are obtained by merging several scans together. On the other hand, in Fig. 2a we limited the profile to the scanner span to avoid repositioning the sample at every temperature and keeping the illumination conditions the same. Depending on the type of measurements, the signal on the sample is normalized either by the signal on metal, or the signal on the sample area, where no SPhP interference is seen or by the peak value.

### Modeling of nano-FTIR spectra

In our simulations of the s-SNOM signal we used an established finite dipole model[51,52], where we adopted the values of the tip radius and the tapping amplitude of 60 nm and 90 nm respectively, as in the experiment. More details are given the Supplementary Note S4. In the case of the LAO/STO samples, we approximated LAO, 2DEG and STO by distinct and spatially homogeneous layers, so that the dielectric function changes sharply at the LAO/2DEG and 2DEG/STO boundaries.

## Data availability

All data that support the findings of this study are available from the corresponding author upon reasonable request.

## Code availability

All code used in this study is available from the corresponding author upon reasonable request.

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

## Acknowledgements

The work of Y.Z., W.L., A.B. and A.B.K. was supported by the Swiss National Science Foundation (grants 200020_185061 and 200020_201096). The work of A.W., M.B., S.G., C.W.R., L.K. and J.M.T. was supported by the Swiss National Science Foundation. The work of W.L. was supported by the National Natural Science Foundation of China (grants 12004196 and 2127803).

## Author contributions

Y.Z., A.B. and W.L. conducted near-field optical measurements, J.L.M.v.M., C.W.R. and J.T. performed far-field optical measurements, A.W., M.B. and S.G. prepared samples and performed transport mea-surements, Y.Z., I.C. and A.B.K. numerically modeled data, J.D., D.v.d.M. and L.K. contributed to theoretical understanding, A.B.K. and J.M.T. conceived the idea of the project. Y.Z. and A.B.K. wrote the manuscript using input from all authors. All authors contributed to discussions.

## Competing interests

The authors declare no competing interests.
