## [Peer Review File · Nature Communications]

Thermal and electrostatic tuning of surface phonon-polaritons in LaAlO₃/SrTiO₃ heterostructuresREVIEWER COMMENTS

Reviewer #1 (Remarks to the Author):

The authors employ broadband, cryogenic, infrared nanospectroscopy to study the higher-frequency SPhP resonance in STO and LAO/STO heterostructures. Their experimental studies are challenging but experiments are performed carefully and systematically. The temperature dependence and gating dependence of the 2D gas at the interface of LAO/STO is also studied via its effect on the STO SPhP resonance onset. The topic is of immense current interest and this impressive work contains sufficient new data and results to be published in Nature Communications. There are a few scientific questions that the authors should address so as to increase the impact of this work.

1. The authors use only ω_{onset} for analysis of temperature-dependence of the 2D electron gas. However, the peak position of the SPhP resonance in LAO/STO is also temperature dependent and it appears to the eye that its temperature dependence is different compared to that seen for the SPhP on surface of bulk STO. The authors should comment and/or address this point.

2. The simulations using the extended (or finite) dipole model in this work, although acceptable, sometimes underestimate and sometimes overestimate the scattering amplitude compared to experiment. It is well known that the parameters in this model are probably frequency dependent and therefore this model is more challenging to implement for broadband spectra. The authors should comment on this matter. Also, have the authors considered numerical simulations that have recently appeared in the literature and that do a better job of calculating the near-field SPhP spectra?

3. Do the near-field phase spectra provide additional information not present in the amplitude spectra?

4. There should be more discussion why the gating only works well for negative voltages. Also, a discussion comparing the temperature dependence and gating dependence of the 2D electron gas would be useful. The "Discussion" section of the manuscript appears short and a bit rushed, so this can be improved.

Reviewer #3 (Remarks to the Author):

The authors report a thorough study on the tunability of the surface phonon polaritons (SPhPs) in SrTiO₃ (STO) systems. With the state-of-the-art cryogenic s-SNOM platform equipped with nano-FTIR, the SPhPs propagating about hundred microns long are clearly revealed. Moreover, the authors show that the SPhPs in pristine STO are hardly tunable, while the onset frequency of the SPhPs in LAO/STO heterostructure hosting interfacial 2DEG is both temperature and static sensitive. The authors ascribe the prominent tunability of the SPhPs in LAO/STO to the temperature-dependent of polaronic screening of the charge carriers and the gate-induced change of the dielectric function of the 2DEG. In my point of view, the main result, that the SPhPs in LAO/STO can be tuned by both the temperature and the gate voltage, is not that surprising but meaningful. Overall, their data seem technically sound, the results are convincing and the manuscript is well-written. Hence, I think this manuscript can be considered for Nature Communications, provided the authors

can address the following questions appropriately.

1. The STO substrate used in this work is (001) oriented with C4 symmetry, why the propagation lengths of SPhPs are different at two orthogonal metal edges. Whether the direction of the step and terrace should be taken in account for the anisotropic SPhP propagating?
2. In addition to the onset frequency, how do other parameters of SPhP (for example, fringe spacing $\langle d \rangle$, decay length $\langle L \rangle$) involve with temperature and gate voltage?
3. Why both second and third harmonics of the s-SNOM amplitude are measured for pristine STO but only the third harmonic signal is adopted for LAO/STO sample?
4. In Figure S5, the real-space nano-spectroscopy of SPhPs in LAO/STO shows near-field amplitude maxima at some specific frequencies, which is very different from that in pristine STO (see Figure 1). Please elucidate the underlying physics.
5. I think the ϕ in the damped-sine function (Page 4) for fitting the spatial profile of near-field amplitude is not the same as that in equation (1) and (2). The former is the phase for damped-sine function fitting, while the latter is the angle between the SPhP momentum and the x-axis (Page 5).

Dear Editor,

We thank both Reviewers for their thoughtful and constructive reports. We are very glad to see that both of them are positive about the publication of our manuscript. Below we address all their remarks and criticisms.

Reply to Reviewer 1

1. The authors use only ω_{onset} for analysis of temperature-dependence of the 2D electron gas. However, the peak position of the SPhP resonance in LAO/STO is also temperature dependent and it appears to the eye that its temperature dependence is different compared to that seen for the SPhP on surface of bulk STO. The authors should comment and/or address this point.

We fully agree that the peak position may be a useful observable for this analysis. Unfortunately, we are working at the edge of the spectral range of the broadband nano-FTIR laser, and the data become progressively noisier at low frequencies. Therefore, the determination of the onset frequency is more accurate than the determination of the peak frequency. For the Reviewer's reference, in Fig. R1 we present the experimentally determined and simulated peak frequency for LAO/STO and STO as a function of temperature. It is clear that while the overall trend of increasing the peak frequency at cooling down is present in all curves (experimental and simulated), it is quite difficult to make any statement about the different behavior of LAO/STO and STO, given the big error bars.

Fig. R1. The peak frequency as a function of temperature. Black circles and squares are respectively experimental and model values on pristine STO, extracted from Fig. 2d and 2e. Red downward and upward triangles are the experimental and simulated values obtained on the LAO/STO system.

Another reason for sticking to the onset frequency in the manuscript is that the procedure used to determine ω_{onset} is robust with respect to the possible presence of a frequency dependent instrumental scaling factor, while the peak frequency can be easily shifted by a multiplication by a non-constant spectral background. Apart from that, the onset frequency can be logically associated with the upper SPhP frequency, (as discussed in the manuscript), while the peak frequency results

from the interplay of many factors, which involve both the sample and the instrumental (i.e. tip) properties.

For all these reasons we decided to focus on ω_{onset} .

A sentence is added to the main text: “We note that the physical meaning of the peak frequency and interpretation of its temperature dependence are less clear than the ones of the onset frequency.”

2.The simulations using the extended (or finite) dipole model in this work, although acceptable, sometimes underestimate and sometimes overestimate the scattering amplitude compared to experiment. It is well known that the parameters in this model are probably frequency dependent and therefore this model is more challenging to implement for broadband spectra. The authors should comment on this matter. Also, have the authors considered numerical simulations that have recently appeared in the literature and that do a better job of calculating the near-field SPhP spectra?

Indeed, quantitative modelling of the broad-band SNOM spectra is a great challenge, and many approaches have been proposed in the literature to tackle the issue. We have chosen the finite-dipole model [A. Cvitkovic, N. Ocelic, R. Hillenbrand, Opt. Express 15, 8550 (2007); S. Amarie, F. Keilmann, Phys. Rev. B 83, 045404 (2011); B. Hauer, A.P. Engelhardt, T. Taubner, Opt. Express 20, 13173 (2012)], which takes into account the elongated shape of the tip and is therefore a major step forward with respect to the basic point-dipole model. The finite dipole has been shown to work well for the nano-FTIR spectra in polar materials with strong Reststrahlen bands. Specifically, in STO and LAO/STO this model was successfully applied in Ref. [J. Barnett et al, Adv. Funct. Mater., 30, 2004767 (2020)]. As the Referee correctly points out, the agreement is acceptable, but not perfect. In particular, it is difficult to reproduce the absolute amplitude (which is also the case in the previous publication of Barnett et al, see their Fig.3). Knowing this, we do not draw any conclusions from the absolute amplitude.

We are aware of more recent modelling approaches, for example [A. McLeod et al, Phys. Rev. B 90, 085136 (2014); B.-Y. Jiang et al. Journal of Applied Physics 119, 054305 (2016); S. T. Chui et al, Phys. Rev. B 97, 081406(R) (2018)]. However, our impression is that these methods, which are mathematically and computationally quite involved, need to pass further tests in several groups of nano-FTIR practitioners. Therefore, we feel more confident using the finite-dipole model, which is, albeit not free of drawbacks, is widely used by the community and which can be reproduced by other groups working on this subject.

3.Do the near-field phase spectra provide additional information not present in the amplitude spectra?

The near-field phase spectra were measured by us as well (see Fig.R2). They are more featureless than the amplitude spectra and moreover, the phase becomes undefined where the amplitude is close to zero (above 730 cm^{-1}). Therefore, we could not draw any essential additional information from the phase spectra as compared to the amplitude spectra.

Fig. R2. Metal-normalized nano-FTIR amplitude (solid lines) and phase (dash-dotted lines) spectra of STO at four temperatures.

4. There should be more discussion why the gating only works well for negative voltages.

The reasons for a low gating efficiency at positive voltages in our combined SNOM-transport experiments are not entirely clear. We tested the same devices in a standard transport setup, and they showed a normal operation at both signs of gating. We can only speculate that the problem arises from the (unavoidable) presence of high-frequency radiation with a photodoping effect in the cryo-SNOM chamber. In the revised manuscript, we have made the corresponding sentence more specific.

Also, a discussion comparing the temperature dependence and gating dependence of the 2D electron gas would be useful. The “Discussion” section of the manuscript appears short and a bit rushed, so this can be improved.

Following the nice suggestion of the Referee, we added discussion comparing the temperature and gating dependence of the 2D gas into the Discussion.

Reply to Reviewer 3

1. The STO substrate used in this work is (001) oriented with C4 symmetry, why the propagation lengths of SPhPs are different at two orthogonal metal edges. Whether the direction of the step and terrace should be taken in account for the anisotropic SPhP propagating?

Even though the difference between the propagation lengths for the two edge orientations is at first surprising, we prefer not to attribute it to a sample anisotropy. As we mentioned in the text, the decay length obtained by the damped-sine function fitting is artificially suppressed because of the limited size of the illumination spot and can therefore serve only a lower bound for the true SPhP propagation length. The two measurements were done at different cooldowns using different AFM tips. After the tip exchange, the parabolic mirror is always realigned, which may change the shape

and position of the focal spot. It is not excluded that the difference between the two values of L_{exp} is caused by variations of the optical configuration.

2. In addition to the onset frequency, how do other parameters of SPhP (for example, fringe spacing d_{exp} , decay length L_{exp}) evolve with temperature and gate voltage?

This is a very good point! While it was one of our main objectives to detect the evolution of the fringe spacing and decay length with temperature (both in STO and LAO/STO) and gate voltage (in LAO/STO), we could not, unfortunately, observe a significant effect as a function of any of these parameters. For pristine STO, $d_{\text{exp}}(T)$, and $L_{\text{exp}}(T)$ are shown in Fig. 2b. d_{exp} weakly increases with cooling down, as expected, due to a slow temperature change of the real part of the dielectric function of STO (Fig. 2c). As has been mentioned in the main text, the low value and the slow change of L_{exp} do not compare well with the theoretical expectations of the propagation length (based on the T dependence of the imaginary part of epsilon, Fig. 2c), which is likely because our experiment provides only a lower bound for the true SPhP propagation length (see also the reply to point 1 of the Referee).

For LAO/STO, we observed roughly the same trends in $d_{\text{exp}}(T)$, and $L_{\text{exp}}(T)$ as in STO and we decided, because of the lack of space, not to dwell on these results.

We did a series of the SPhP interferometry measurements on LAO/STO for different gate voltages and the dependence of d_{exp} and L_{exp} on V_G is weak, if any. In Fig. R3 a-c we show three $(\omega-x)$ maps measured at $V_G = 150$ V, 0 V and -150 V. The difference between the spatial signal profiles is hardly noticeable, as one can also see in Fig. R3d. The absence of a strong gate voltage effect on this interference pattern is explained by the fact that the fringes are determined by low-momentum SPhPs. The electric field for these SPhPs penetrates deep into the STO substrate and therefore is weakly affected by gate-induced changes in 2DEG. The much higher effect of the gate voltage on ω_{onset} is because it is determined by high-momentum SPhPs, which are more localized to the surface and therefore more sensitive to the 2DEG. Quantitatively, this is demonstrated in Fig. R4, where we present simulations of the surface-polariton dispersion at 0 V and at -150 V, using the same assumptions as for calculations shown in Fig. 4d.

As many readers may ask the same question, we decided to add these two figures (Fig. R3 and Fig. R4) into the Supplementary information (Fig. S9 and Fig. S10), and we added a corresponding text into the Discussion section. This also addresses the criticism of the 1st Referee regarding the 'rushed' style of the Discussion section.

Fig. R3. a-c, Hyperspectral (ω - x) maps of the SNOM amplitude in LAO/STO at 10 K at $V_G=+150$ V, 0V and -150 V respectively. **d**, The spatial profiles at different voltages at $\omega=690$ cm^{-1} . The data are normalized to the reference signal far from the edge.

Figure R4. **a** and **b**, Simulated SPHP – plasmon polariton dispersion in the LAO/STO system with a 2DEG at two gate voltages: 0 (**a**) and -150 V (**b**). Applying gate voltage was mimicked by adding 24 to the real part of $\epsilon_{2\text{DEG}}(\omega)$. The solid lines denote the SPHP branch, the dashed lines – the 2DEG plasmon polariton branch, determined as maxima of $\text{Im}[r_p(\omega)]$ as function of ω for a fixed value of q . In panel **c**, the two branches are shown together for the two values of the gate voltage. Region A corresponds to the low-momentum SPHPs, which determine the interference patterns in the hyperspectral (ω - x) maps (Fig. R3). Region B corresponds to the high-momentum SPHPs, which define the onset frequency seen in the nano-FTIR spectra. Region C corresponds to the 2DEG plasmon-polariton branch, studied by us in Ref. [W. Luo et al, Nature Communications **10**, 1-8 (2019)].

3. Why both second and third harmonics of the s-SNOM amplitude are measured for pristine STO but only the third harmonic signal is adopt for LAO/STO sample?

In fact, we use both harmonics for both samples. As a rule, we use the 2nd harmonics for the spatial scans (Fig. 1 and Fig. S2 for STO and Fig. S5 for LAO/STO) and the 3rd harmonics for the nano-FTIR spectra (Fig. 2d for STO and Fig. 3a for LAO/STO). The reason for using different harmonics in these

two cases is a compromise between the signal to noise ratio (which is more essential in the spatial scans) and the relative amount of the near field response as compared to the far-field background (which is more important for spectroscopy as it is frequency dependent).

4. In Figure S5, the real-space nano-spectroscopy of SPhPs in LAO/STO shows near-field amplitude maxima at some specific frequencies, which is very different from that in pristine STO (see Figure 1). Please elucidate the underlying physics.

We thank the Referee for pointing this out. The sharp maxima seen at specific frequencies in Fig. S5 come from the normalization on the metal used as a reference. In this specific measurement, the nano-FTIR spectrum on the metal, $s_{2,\text{met}}$, showed minima at some frequencies, which resulted in artificial maxima in the ratio $s_2/s_{2,\text{met}}$. We agree that this creates confusion, even though the goal of Fig.S5 – showing the SPhP oscillations – is achieved. In the new version of the supplementary Fig.S5 (see also Fig.R5 in this reply), we normalize the same data by the signal far from the edge, where the artificial maxima are absent, but the SPhP fringes are still nicely seen.

Fig. R5. Hyperspectral (ω - x) map of LAO/STO showing the near-field amplitude $s_2/s_{2,\text{ref}}$ normalized to the signal far from the edge, as a function of the distance x between tip and metal (the same as the new Fig. S5).

5. I think the φ in the damped-sine function (Page 4) for fitting the spatial profile of near-field amplitude is not the same as that in equation (1) and (2). The former is the phase for damped-sine function fitting, while the latter is the angle between the SPhP momentum and the x-axis (Page 5).

Absolutely! We are sorry for the confusion. The problem is fixed by replacing φ with φ_0 in the damped-sine formula.

Remark to both Reviewers

Having addressed the Reviewer's remarks, we would like to ask their permission to correct, by our own initiative, a paragraph in the original manuscript, where we found and corrected a mistake. It concerns the last part of the last paragraph of the Results section "If we assume that this change is caused solely by the 2DEG then, using the Drude model, we can relate $\delta\varepsilon_1(V_G)$ to the change of the carrier density $\delta n_{2\text{DEG}}(V_G) = n_{2\text{DEG}}(V_G) - n_{2\text{DEG}}(0)$: $\delta\varepsilon_1(V_G) = -(1/t) \cdot \delta n_{2\text{DEG}}(V_G) \cdot (e^2/\varepsilon_0 m^* \omega^2)$, where m^* is the effective mass of the charge carriers and ε_0 is the vacuum permittivity. Assuming that m^* is 3.2 times the free-electron mass⁵⁴ and setting ω to 740 cm^{-1} (the onset frequency) we obtain $\delta n_{2\text{DEG}} = -5.7 \times 10^{13}$ and $-1.15 \times 10^{14} \text{ cm}^{-2}$ for $V_G = -50$ and -150 V respectively. These values are reasonable and consistent with the reported gate voltage efficiency of $\sim 10^{11} \text{ cm}^{-2}/\text{V}$ in similar devices³⁵. ». The extracted values of $\delta n_{2\text{DEG}}$ assume that the simple Drude model applies not only to the DC transport, but also to the optical conductivity in the mid-infrared range without modifications. In fact, these values are higher than what is expected from the gate efficiency in Ref.35, contrary to what has been written earlier. Therefore, we do not use this argument anymore and we carefully thought of a different one.

It is important that the real part of the dielectric function of the 2DEG increases with applying a negative gate voltage (which is a solid experimental fact independently of interpretations), which means that its absolute value is decreasing. This agrees qualitatively that the carrier density decreases at negative voltage. Furthermore, the resulting value of $\text{Re}[\varepsilon_{2\text{DEG}}(\omega_{\text{onset}}, V_G)]$ is still negative even at the strongest voltages applied, which means that the 2DEG rests metallic, according to the transport measurements. We modified the corresponding piece of text to the following: "Keeping in mind that $\text{Re}[\varepsilon_{\text{STNO}}(\omega_{\text{onset}})] \approx -40$ (Fig.3b), these changes imply that $\text{Re}[\varepsilon_{2\text{DEG}}(\omega_{\text{onset}}, V_G = -50\text{V})] \approx -28$ and $\text{Re}[\varepsilon_{2\text{DEG}}(\omega_{\text{onset}}, V_G = -150\text{V})] \approx -16$. First, the lowering of the absolute value of ε_1 is consistent with the expected decrease of the carrier density at negative voltages. Second, the value remains negative, indicating the presence of carriers even at the highest absolute voltage applied. Therefore, we conclude that the hypothesis of the connection between the density of the carriers and the shift of the onset frequency is reasonable. ».

A more complicated connection between the changes of the transport value of $n_{2\text{DEG}}$ and the changes of the optical dielectric function at ω_{onset} (which is determined not only by the Drude peak but also the mid-infrared polaronic band) then just the simple Drude equation in the cited paragraph is a possible reason of the inapplicability of the latter in our case.

Therefore, we softened the interpretation and replaced the sentence “Below we argue that the origin of this connection is the gate-induced change of the dielectric function of the 2DEG.” with “Below we check numerically if the gate-induced change of the dielectric function of the 2DEG close to the SPhP onset frequency (740 cm^{-1}) may explain this connection.”

REVIEWERS' COMMENTS

Reviewer #1 (Remarks to the Author):

The authors have addressed most of my comments, and the manuscript has been improved. I recommend publication in Nature Communications.

Reviewer #3 (Remarks to the Author):

The authors have provided detailed responses to all of my comments. I think the current version is suitable for publication in Nature Communications.